# Influencing Factors of Delayed Diagnosis of COVID-19 in Gangwon, South Korea

**DOI:** 10.3390/ijerph21050641

**Published:** 2024-05-17

**Authors:** Minhye Park, Seungmin Jeong, Yangjun Park, Saerom Kim, Yeojin Kim, Eunmi Kim, So Yeon Kong

**Affiliations:** 1Infectious Disease Control Division, Gangwon State Government, Chuncheon 25425, Gangwon, Republic of Korea; maniya817@korea.kr; 2Gangwon State Center for Infectious Diseases (Affiliated to Korea Disease Control and Prevention Agency and, Gangwon State Government), Chuncheon 24280, Gangwon, Republic of Korea; onetwopyj@gwcid.or.kr (Y.P.); srkim@gwcid.or.kr (S.K.); 33dear@gwcid.or.kr (Y.K.); emkim@gwcid.or.kr (E.K.); 3Department of Preventive Medicine, Kangwon National University Hospital, Chuncheon 24289, Gangwon, Republic of Korea; 4Strategic Research, Laerdal Medical, 4002 Stavanger, Norway; joyce.kong@laerdal.com

**Keywords:** COVID-19, COVID-19 test, delayed diagnosis

## Abstract

This study aimed to identify the time to diagnosis among COVID-19 patients and factors associated with delayed diagnosis (DD). Data from COVID-19 patients in Gangwon, South Korea, diagnosed between 22 February 2020 and 29 January 2022, were analyzed, excluding asymptomatic cases and those who underwent mandatory testing. DD was defined as a period exceeding 2 or more days from symptom recognition to COVID-19 diagnosis. Univariate analysis was performed to investigate the demographic characteristics, COVID-19 symptoms, and underlying medical conditions associated with DD, followed by multivariate logistic regression analysis for significant variables. Among 2683 patients, 584 (21.8%) were diagnosed within a day of symptom onset. DD rates were lower in patients with febrile symptoms but higher among those with cough, myalgia, or anosmia/ageusia. High-risk underlying medical conditions were not significantly associated with DD. Older age groups, the Wonju medical service area, time of diagnosis between November 2020 and July 2021, symptom onset on nonworkdays, and individuals in nonwhite collar sectors were significantly associated with increased DD risks. These findings were consistent in the sensitivity analysis. This study underscores the need for enhanced promotion and system adjustments to ensure prompt testing upon symptom recognition.

## 1. Introduction

The coronavirus disease 2019 (COVID-19) escalated into a global pandemic, leading to a substantial number of confirmed cases and deaths. A pivotal factor in the increase in confirmed cases before its official designation as a pandemic was COVID-19’s rapid spread as a respiratory infectious disease [1]. Early-stage COVID-19 patients are known for their ability to transmit the virus despite exhibiting mild symptoms [2]. It is estimated that, under normal circumstances, an infected individual may transmit the virus to an average of 2–4 others [3]. Therefore, early diagnosis of symptomatic COVID-19 cases has been recognized as imperative to mitigate its spread. Moreover, timely diagnosis of COVID-19 is essential for expedited patient management and improved health outcomes. Notably, previous cohort study findings have shown that even after adjusting for various confounding variables, a delay of more than five days from symptom onset to confirmed diagnosis was associated with a 69% increase in the likelihood of patients progressing to severe illness [4]. In response, the World Health Organization (WHO) and national governments worldwide advocated for and promoted rapid testing for symptomatic individuals. However, the testing process encountered several obstacles. Some studies reported factors such as the cost of testing, health literacy, trust in the healthcare system, and the accessibility of testing sites as significant barriers [5,6,7]. Other studies reported that some people may have been reluctant to get tested because of privacy concerns and testing positive leading to stigmatization, including workplace discrimination [7,8]. Additionally, other studies reported the potential economic impact of post-test isolation and the fear of disease transmission within testing facilities as reasons for reluctance [7].

While the Korean government advised rapid testing following symptom onset, data on adherence to these guidelines are lacking [9]. Additionally, in order to prepare for future pandemics, it is necessary to determine what factors impede rapid testing after symptoms.

Our study used data from COVID-19 epidemiological investigation reports containing records of symptom onset and diagnosis dates to identify the time to diagnosis (TTD) and which characteristics were associated with delayed diagnosis (DD) of COVID-19.

## 2. Materials and Methods

### 2.1. Setting

Gangwon-do (hereinafter referred to as “Gangwon”) is a special autonomous administrative region in South Korea, covering an area of 16,829.7 km^2^, 94% of which is mountainous. Its total population is 1,528,635, with a population density of 91.3 people per km^2^. This province is characterized by a low population density that is dispersed across a broad area composed of urban and agricultural lands. Notably, over 72% of the population is concentrated in urban areas situated on flat land [10]. Gangwon is composed of 18 Si (cities) and Gun (counties), each hosting a public health center. Public health centers, established with the purposes of optimizing health administration and effectively promoting health policies, serve as pivotal institutions within the cities and counties of Korea [11]. They also play a crucial role in local health initiatives, encompassing functions of both public health and primary care, with a notable emphasis on the implementation of public health management strategies [12,13]. In specific regions characterized by limited medical resources, public health centers function as comprehensive healthcare facilities, equipped with both outpatient and inpatient treatment capabilities, alongside their local public health management functions [14]. Among these 18 public health centers in Gangwon, two are designated as public medical centers, providing essential emergency care and medical services.

The 2015 Middle East Respiratory Syndrome outbreak left many lessons the healthcare system in South Korea needed to learn to respond to novel infectious disease outbreaks, leading to the amendment of the Infectious Disease Control and Prevention Act within the same year. This experience prepared the central and local governments to implement proactive measures at the early stages of the COVID-19 outbreak, such as adopting the 3T (test, trace, and treat) strategy and emphasizing the rapid identification, isolation, and treatment of confirmed COVID-19 cases [15]. Starting 7 February 2020, individuals with suspected COVID-19 symptoms or those who had been in contact with a confirmed patient were eligible to receive a real-time reverse transcription polymerase chain reaction (RT-PCR) test free of charge at designated screening booths in public health centers or hospitals within their cities or counties. Swabs obtained were conveyed to the Gangwon State Institute of Health and Environment or private research institutes within Gangwon, for RNA extraction and subsequent RT-PCR analysis. Commencing from 24 January 2020 and continuing throughout the study period, the Gangwon State Institute of Health and Environment maintained a 24 h emergency response system, enabling uninterrupted COVID-19 diagnostic testing. RT-PCR-based COVID-19 testing operated under the guiding principle of analyzing specimens on the day of collection to ensure expedited results. Consequently, most specimens underwent immediate transportation to the laboratory upon collection [16]. If a real-time RT-PCR test yielded a positive result, the information was forwarded to the public health center of the respective county. The centers checked the information and conducted basic and in-depth epidemiological investigations on all confirmed cases as quickly as possible via phone calls or in-person visits. The basic epidemiological investigation included demographic information, COVID-19 symptoms, the timing of symptom onset, and the reason for the RT-PCR test. The investigation results were registered in the COVID-19 Data Management System of the Korea Disease Control and Prevention Agency (KDCA) in real time. Moreover, public health centers conducted in-depth epidemiological investigations to identify people and locations that the confirmed patient had been in contact with from two days before symptom onset until the RT-PCR test. The findings from these investigations were used to trace the route of transmission and classify contacts to take appropriate measures, including isolation orders. All individuals identified as having been in contact with a confirmed patient were required to undergo a COVID-19 test and were subject to self-isolation or active monitoring, based on the degree of contact. According to the Infectious Disease Control and Prevention Act, those under epidemiological investigation could not refuse cooperation without a valid reason and were required to actively participate in treatment and self-isolation protocols [9]. Furthermore, all travelers entering Korea from abroad were subjected to a COVID-19 test within one day of arrival and a mandatory self-isolation for a specified period. Those in self-isolation were required to take a test immediately if they developed clinical symptoms of COVID-19, and even asymptomatic individuals were required to undergo a RT-PCR test before the end of their self-isolation to verify a negative result prior to being released. The government provided financial support for living expenses or paid leave during the self-isolation period.

### 2.2. Data Source

This study utilized the COVID-19 epidemiological investigation database of Gangwon as the data source. This database comprises both basic and in-depth epidemiological investigation information for confirmed COVID-19 cases based on RT-PCR results from the first reported case within Gangwon. This data source includes demographic characteristics of confirmed patients (name, sex, age, date of birth, occupation, and place of residence), clinical symptoms (presence of symptoms, timing of symptom onset, and type of symptoms), the date of COVID-19 diagnosis, reasons for undergoing testing, cycle threshold (Ct) values, vaccination details (vaccination status, number of doses received, and vaccine type), and information pertinent to the confirmed cases (movement route, relationship with the antecedent confirmed case, and place and time suspected of transmission) [10].

### 2.3. Study Population

The study population included individuals who voluntarily underwent COVID-19 testing upon recognition of symptoms among all confirmed cases reported in Gangwon between 22 February 2020 and 29 January 2022. Individuals who underwent mandatory testing in cases classified as “imported case from abroad” or “confirmed case resulting from contact with a confirmed patient based on epidemiological investigation results” were excluded from the study. Similarly, individuals who were tested during or just before release from self-isolation as mandated by COVID-19 containment policies and asymptomatic individuals who voluntarily underwent testing were excluded.

### 2.4. Variables

The outcome variable was the DD of COVID-19, calculated as the time to diagnosis (TTD), i.e., the duration in days from symptom recognition to the date of diagnosis. Because our data source not including the actual testing dates, calculations were made using the diagnosis dates, under the assumption that diagnosis and COVID-19 testing occurred closely together. To validate this operational definition, we cross-referenced the dates of specimen collection and the dates of diagnosis as recorded in the KDCA COVID-19 Data Management System. An analysis of the interval between the test date (the date of specimen collection) and the diagnosis date (the date recorded in the epidemiological investigation report) for the period from 1 February 2020 to 31 January 2022, indicated that a vast majority of the 16,090 confirmed case reports showed diagnosis within 0 days (28.83%) and 1 day (69.25%) from testing. In other words, since approximately 98% of the cases were diagnosed within 24 h of testing, the gap between the time of testing and that of diagnosis was determined to be negligible. Accordingly, the validity of the operational definition was affirmed, allowing DD to be used as the outcome variable in the main analysis, acting as a surrogate measure for delayed testing. Considering the WHO and governmental recommendations for immediate testing upon symptom onset, testing on the day symptoms appear would be ideal. However, when considering the time required for the transportation of COVID-19 test samples and the typical duration of approximately six hours for COVID-19 RT-PCR testing, samples collected during late evenings or at night are more likely to yield results the following day [17]. Additionally, since the majority of cases had a TTD within 1 day, classifying timely testing as a TTD of 0 or 1 day, and a TTD of 2 or more days as DD, was deemed appropriate to minimize the potential for information bias.

The explanatory variables include sex, occupational groups, medical service areas, time of testing, and symptom recognition on workdays or nonworkdays. The occupational groups were classified based on the International Standard Classification of Occupations: (i) white collar (managers, professionals, technicians, and associate professionals), (ii) pink collar (clerical support workers and service and sales workers), (iii) blue collar (skilled agricultural, forestry, and fishery workers, craft related trades workers, plant and machine operators and assemblers, elementary occupations, and armed forces occupations), (iv) students, and (v) economically inactive population (preschool children, unemployed, and unspecified) [18]. Age was divided into the following age groups: 0–19, 20–39, 40–64, and ≥65 years.

The region was categorized based on the medical service areas classified in the local health improvement plan proposed by the government in 2019. Medical service areas were determined not by geographical location but by population size, health service utilization, and living areas [19]. There are six medical service areas within Gangwon, as follows: Chuncheon area (including Chuncheon-si; Hongcheon, Hwancheon, Yanggu, and Cholwon counties), Wonju area (including Wonju-si and Hoengseong county), Yeongwol area (including Yeongwol, Jeongseong, and Pyeongchang counties), Gangneung area (Gangneung-si), Donghae area (including Donghai-si, Taebaek-si, and Samcheok-si), and Sokcho area (including Sokcho-si, Goseong, Yangyang, and Inje counties). Among these six medical service areas, the Chuncheon and Wonju areas include cities with populations of over 200,000 (Chuncheon-si and Wonju-si) as part of the medical service area, while the Gangneung area consists of a single city with a population of over 200,000. The other areas are composed of cities and counties with populations under 100,000.

The time of testing was categorized based on epidemic phases as classified in the KDCA COVID-19 epidemiological reports. Phase 1, spanning from 20 January 2020 to 12 November 2020, marks the period from the first confirmed COVID-19 case to the first wave, with mass outbreaks nationwide and the subsequent second wave when outbreaks associated with religious facilities and urban gatherings in the Seoul Capital Area occurred. Phase 2, spanning 13 November 2020 to 6 July 2021, encompassed the third wave, featuring nationwide outbreaks in correctional facilities, hospitals, nursing homes, and religious facilities. Similarly, Phase 3 covered 7 July 2021 to 29 January 2022, marked by the Delta variant’s spread, leading to a rise in cluster outbreaks and contact-related small-group or individual infections. In Phase 4, beginning in late January 2022, Omicron became the dominant variant, and the spread of infection increased as the reproduction number increased to over 1 [20].

Symptom recognition days were distinguished between workdays and nonworkdays (i.e., weekends and public holidays). Vaccination status was classified as unvaccinated (having received no COVID-19 vaccine), partially vaccinated (having received the first dose of the COVID-19 vaccine or being within 14 days postvaccination), and fully vaccinated (having completed the recommended number of vaccine doses plus 14 days). COVID-19 vaccination in Korea started on 26 February 2021. Clinical symptoms of COVID-19 were categorized into fever, chills, cough, sputum, difficulty breathing, chest pain, loss of consciousness, cyanosis, sore throat, headache, myalgia, runny nose and nasal congestion, fatigue, diarrhea, vomiting, anosmia/ageusia, abdominal pain, dizziness, loss of appetite (anorexia), and others. A number of participants reported two or more symptoms. Underlying medical conditions were distinguished between conditions associated with high risk for severe COVID-19 and non-high-risk conditions, as defined by the US Centers for Disease Control and Prevention [21]. High-risk conditions include diabetes, cancer, kidney disease requiring dialysis, heart disease, cerebrovascular disease, asthma, chronic lung disease, liver disease (cirrhosis, fatty liver, and autoimmune diseases), mental illness (mood disorders and schizophrenia), dementia, physical inactivity, and the use of immunosuppressant medication. Conditions such as pregnancy, obesity, and smoking were also considered high risk, though not verifiable. Multiple responses were accepted from the patients.

### 2.5. Data Analysis

The basic epidemiological characteristics were summarized using frequencies and percentages and analyzed using the chi-square test. Frequencies and percentages were also calculated for each symptom and underlying medical condition, and these factors were analyzed using the chi-square test. If the expected frequency assumption for the chi-square test was not met, Fisher’s exact test was performed. Given the large number of symptoms and underlying medical conditions analyzed using the chi-square test, the Bonferroni correction was applied with adjusted alpha levels of 0.0025 (0.05/20) for symptoms and 0.0033 (0.05/15) for underlying medical conditions, indicating statistical significance.

With DD defined as a period exceeding 2 or more days from symptom recognition to COVID-19 diagnosis, multivariate logistic regression analysis was performed to examine the association of DD with the variables, including those that showed a *p*-value < 0.2 in the univariate analysis, namely age, the medical service areas, time of diagnosis, symptom recognition on workdays vs. nonworkdays, occupational groups, and underlying medical conditions. Additionally, sensitivity analysis was performed by adjusting the definition of DD as a TTD exceeding 2, 3, and 4 days, and the results were compared with the main result. Statistical analysis was performed using SAS 9.4. (SAS Institute, Cary, NC, USA).

### 2.6. Ethical Considerations

This study was approved by the Institutional Review Board of Kangwon National University Hospital (IRB approval number: KNUH-2021-02-001). Informed consent was waived because of the retrospective nature of this study. This study was performed in accordance with the principles of the Declaration of Helsinki.

## 3. Results

Between 22 February 2020 (the date of the first confirmed case) and 29 January 2022, Gangwon recorded 15,704 confirmed COVID-19 cases. Of these, 3175 cases were identified as individuals who voluntarily underwent testing, after excluding 374 imported cases and 12,155 cases in which the individual was tested due to being classified as a contact of a confirmed patient based on epidemiological investigation or tested during or just before release from isolation. After excluding an additional 486 individuals who reported no symptoms before testing and five cases with unrecorded symptom onset dates, a total of 2683 cases remained eligible for analysis. Table 1 shows the distribution of TTD. Among these, 584 cases (21.8%) had a TTD of within 1 day, 764 cases (28.5%) had a TTD of 2 days, and 495 cases (18.4%) had a TTD of 3 days. Additionally, 183 cases (6.8%) had TTD of 8 or more days.

Table 2 presents the epidemiological distribution of variables regarding TTD within 1 day and a TTD of 2 days or more (indicating DD). The ≥65 years age group had the highest DD rate at 84.9%. Meanwhile, there was no statistically significant difference in DD according to sex (*p* = 0.44), whereas a statistically significant difference according to region was found (*p* = 0.01). With respect to the time of diagnosis, phase 2 showed the highest percentage of DD (84.8%). DD was more common for cases with symptom onset occurring on weekends or public holidays (82.4%) than on workdays (76.7%). With respect to occupational groups, DD rates were as follows: 74.1% for white-collar workers, 84.1% for pink-collar workers, 70.9% for blue-collar workers, 80.1% for students, and 77.2% for the economically inactive population. No significant difference in DD according to vaccination status was found (*p* = 0.88).

Table 3 shows the distribution of how DD status varied according to symptoms. Symptoms identified as statistically significant based on a *p*-value of 0.0025 included fever, cough, myalgia, and anosmia/ageusia. In comparison to the entire study population, which exhibited a DD rate of 78.2%, the group presenting with fever showed a lower DD rate of 70.8%. Conversely, the remaining statistically significant symptoms were associated with higher DD.

Table 4 shows the distribution of how DD status varied according to underlying medical conditions. While the entire study population showed a DD rate of 78.2%, the group with underlying medical conditions showed a DD rate of 82.9%, indicating a significant increase in DD in individuals with underlying medical conditions. However, high-risk underlying medical conditions did not show a statistically significant correlation with DD. Specifically, hypertension and dyslipidemia were identified to be conditions with significant influence on DD (*p* = 0.0033).

Multivariate logistic regression analysis was performed with the inclusion of variables with a *p*-value ≤ 0.2 in univariate analysis (Table 2), as follows: age, medical service area, time of diagnosis, symptom recognition on workdays vs. nonworkdays, occupational groups, and underlying medical conditions (Table 5). Although the number of symptoms was found to be significant in the chi-square test, it was excluded from the regression analysis due to the greater likelihood of being a consequence of DD, considering the causal relationship. The analysis revealed that, relative to individuals aged 0–19, the risk of DD was significantly higher in the age groups 20–39 (adjusted odds ratio [aOR] 2.03, 95% confidence interval [CI] 1.39–2.95), 40–64 (aOR 3.12, 95% CI 2.04–4.76), and ≥65 years (aOR 3.60, 95% CI 2.21–5.88). In terms of the medical service areas, the Wonju area (aOR 1.38, 95% CI 1.08–1.76) showed a higher risk of DD relative to the Chuncheon area. In terms of the time of diagnosis, the risk of DD was significantly higher in Phase 2 (aOR 1.57, 95% CI 1.14–2.15) relative to Phase 3. The risk of DD was higher if symptom onset occurred on a holiday or weekend (aOR 1.25, 95% CI 1.05–1.49) relative to workdays. Moreover, significant differences in DD were observed across occupational groups, with increased risk for pink-collar workers, blue-collar workers, economically inactive populations, and students, relative to white-collar workers, by 1.84-fold (95% CI 1.33–2.54), 1.43-fold (95% CI 1.07–1.91), 1.92-fold (95% CI 1.28–2.88), and 1.44-fold (95% CI 1.08–1.92), respectively. DD was not significantly associated with underlying medical conditions.

Table A1, Table A2 and Table A3 (Appendix A) present the results of the sensitivity analyses for the multivariate logistic regression analysis. In the sensitivity analyses with DD defined as a TTD of ≥3, ≥4, or ≥5 days, the results showed trends similar to those observed in Table 5. When DD was defined as a TTD of ≥3 days, both the Wonju and Donghae areas exhibited a significantly higher risk of DD relative to the Chuncheon area, the latter serving as the reference. When classified by occupational groups, pink-collar workers, blue-collar workers, and students showed no significant differences relative to white-collar workers. When DD was defined as TTD of ≥4 days, the results showed similar trends as those shown in Table 5, except there were no significant differences in the age groups of 20–39, 40–64, and ≥65 years. No significant differences were observed in the DD rates between symptom onset on workdays and nonworkdays, while pink-collar workers and students showed no significant difference relative to white-collar workers. When DD was defined as a TTD of ≥5 days, a significant difference was observed only in the age group of ≥65 years relative to the 0–19 year age group. In terms of medical service areas, both the Wonju and Gangneung areas showed significantly higher risks of DD relative to the Chuncheon area, while no significant differences were observed between symptom recognition on nonworkdays and workdays. Moreover, among occupational groups, pink-collar workers and students showed no significant difference relative to white-collar workers.

## 4. Discussion

This study investigated the relationship between delayed testing—using DD as a surrogate measure because of the absence of specific testing date data—following the onset of COVID-19 symptoms and the associated characteristics of confirmed cases. Despite the onset of symptoms, about 80% of symptomatic individuals did not receive a COVID-19 diagnosis within a days. Age, time of diagnosis, medical service areas, symptom onset on workdays versus nonworkdays, and occupational group were pinpointed as factors influencing DD.

Despite recommendations from the WHO and governments around the world, including Korea and the USA, to seek immediate COVID-19 testing upon symptom onset, only a small proportion of patients adhered to this guidance [22,23,24]. In Korea, where testing costs were fully subsidized, only 21.8% of the patients in this study underwent testing either on the day of symptom onset or the following day.

The findings in this study showed that the likelihood of DD was higher among older patients, and, in particular, the risk of DD was 2.93 times higher among older adults aged ≥65 years compared to individuals aged 0–19. Other studies have also found that the risk of delayed COVID-19 diagnosis was higher among older adults [25,26,27]. Given older adults aged ≥65 years have an increased risk of disease progression to severe COVID-19, timely diagnosis is particularly crucial for this age group [28]. However, symptom recognition may be delayed in older adults who often view their aches and pains as part of the normal aging process, with COVID-19 symptoms potentially dismissed as such chronic conditions [29]. Consequently, there is a crucial need for targeted outreach and education for older adults, as disease symptoms can manifest nonspecifically in this group.

Among the six medical service areas in Gangwon, the Wonju area was found to have a higher risk of DD compared to other areas. Wonju-si, which is at the center of the Wonju area, is the most populous city in Gangwon, hosting a variety of medical institutions, including university hospitals, medical centers, and public health centers. However, this area has a lower screening clinic-to-population ratio compared to other areas, potentially influencing DD. For instance, while the Chuncheon area, with a population size similar to the Wonju area, had 10 screening clinics, Wonju had only five. Cultural differences across regions might also have played a role in these differences. In the analysis of regions categorized by population size of cities, counties, and districts instead of medical service areas, the results showed no significant differences.

In the analysis based on phases, the results showed that Phase 2 (20 November 2020 to 5 July 2021) had the highest risk of DD. Unlike during Phase 1, free diagnostic testing was available to everyone, irrespective of symptom presentation or epidemiological association, during Phase 2, and as a result, the cost of testing was not a barrier that caused reluctance for testing. Nevertheless, this phase was marked by the first substantial increase in the number of patients, which could have contributed to the observed delays in testing and confirming diagnoses.

The analysis based on a workday/nonworkday distinction revealed that the risk of DD significantly increased when symptoms were recognized on nonworkdays (weekends or public holidays) as opposed to workdays. This finding was consistent with results reported in two previous studies conducted in Singapore and Japan [25,26]. Both studies speculated that the reason might be that because many healthcare facilities are closed on weekends and public holidays, individuals with mild symptoms opted to wait until a weekday to seek testing instead of going to a crowded emergency room. Although numerous screening centers in Korea remained operational during weekends and public holidays, the potential for testing delays still existed due to the closure of some healthcare facilities on nonworkdays.

When analyzed by occupational groups, non-white-collar groups showed a higher risk of DD compared to white-collar groups. This could be attributed to white-collar jobs typically having more flexible working hours and better leave conditions than pink or blue-collar jobs, potentially influencing DD [30]. Public screening centers and healthcare facilities mainly operated from 9 am to 6 pm, with most screening stations operated through emergency rooms being available outside these hours. Being diagnosed with COVID-19 through screening at emergency rooms may prove to be more cumbersome than through dedicated screening centers, posing additional challenges for workers with inflexible work schedules. Moreover, white-collar workers were encouraged to work from home during the COVID-19 pandemic, which made it easier to get tested in a timely manner, as compared to pink or blue-collar workers. A Japanese study found no significant difference in DD between economically inactive individuals and office workers [26]. However, our analysis revealed that economically inactive individuals had a higher risk of DD compared to white-collar workers. This may be due to the fact that concerns about starting an outbreak within schools or workplaces might have prompted students or workers to undergo testing without delay. In Korea, the government’s provision of paid leave and subsidies for those in COVID-19 isolation significantly reduced concerns over economic losses. Instead, the fear of workplace outbreak stigma as the index case became more pronounced.

Previous studies have indicated an association between vaccination status and DD. Analysis has shown that individuals hesitant to vaccinate may perceive COVID-19 as less threatening, consequently leading to delayed testing [25]. In contrast, findings from an interview survey suggested that vaccinated individuals might not seek or postpone testing due to a belief in the protective effect of the vaccination against COVID-19 infection [7]. However, this study found no significant differences in DD between vaccinated and unvaccinated individuals, potentially reflecting the combined influence of these opposing factors.

DD rates showed significant differences based on clinical symptoms. Individuals with febrile symptoms were more inclined to undergo testing within one day of symptom onset, whereas individuals with symptoms such as cough, myalgia, and anosmia/ageusia tended to show a higher DD rate. Fever, a key symptom of COVID-19 widely recognized through media, can be measured easily and accurately, leading those with febrile symptoms to seek testing immediately. Conversely, symptoms resembling those of the common cold, such as cough and myalgia, may have led to postponed testing. Interestingly, despite the fact that anosmia and ageusia were mentioned as symptoms unique to COVID-19, individuals with these symptoms showed a higher rate of DD.

This study had some limitations. First, epidemiological investigations rely on personal statements, and thus, the details regarding the symptoms and their onset dates may not always be accurate. Additionally, epidemiological investigations are delayed when there is a sudden surge in the number of COVID-19 cases due to mass outbreaks, and as a result, the patients may not always be able to accurately remember the time of symptom onset. Second, socioeconomic status, cohabitation status, and COVID-19 infection counts may represent residual confounders, yet these were not verifiable in our data source. Third, given that the analysis in this study was limited to residents of a single province in South Korea, results from countries with distinct public health policies or different cultural characteristics may be different from the findings of this study.

## 5. Conclusions

Many individuals, particularly older adults, those with symptom onset on nonworkdays, and non-white-collar workers, did not undergo timely testing for COVID-19 despite exhibiting symptoms, displaying a higher likelihood of DD. Additionally, the phase of the outbreaks and the area of residence (medical service areas) were found to be associated with DD. Therefore, promotional efforts to ensure timely testing among older adults and in areas known for testing delays are essential while implementing systemic measures to minimize delays in testing and diagnosis across various occupational groups is also needed. Further studies considering other factors, including socioeconomic status, living conditions, and the number of COVID-19 infections, could help us understand the cause of DD. Studies conducted in other countries could also provide evidence for the external validity of our results.

## Figures and Tables

**Table 1 ijerph-21-00641-t001:** Distribution of time to diagnosis.

Time to Diagnosis (Number of Days)	Patients (N)	Patients (%)	Cumulative Ratio (%)
Total	2683	100	
0–1	584	21.8	21.8
2	764	28.5	50.3
3	495	18.4	68.7
4	281	10.5	79.2
5	177	6.6	85.8
6	97	3.6	89.4
7	102	3.8	93.2
≥8	183	6.8	100.0

**Table 2 ijerph-21-00641-t002:** Participants’ general characteristics.

Variable	Total N (%) ^a^	Tested within 24 h of Symptom Onset N% ^b^		DD (≥48 h after Symptom Onset) N% ^b^		*p*-Value
Total number of patients	2683 (100)	584	21.8	2099	78.2	
Age						<0.01
0–19	348 (13)	119	34.2	229	65.8	
20–39	931 (34.7)	230	24.7	701	75.3	
40–64	966 (36)	169	17.5	797	82.5	
≥65	438 (16.3)	66	15.1	372	84.9	
Sex						0.44
Male	1483 (55.3)	331	22.3	1152	77.7	
Female	1200 (44.7)	253	21.1	947	78.9	
Medical service area						<0.01
Chuncheon	904 (33.7)	221	24.4	683	75.6	
Sokcho	360 (13.4)	67	18.6	293	81.4	
Gangneung	359 (13.4)	94	26.2	265	73.8	
Donghae	205 (7.6)	43	21	162	79	
Yeongwol	111 (4.1)	19	17.1	92	82.9	
Wonju	744 (27.7)	140	18.8	604	81.2	
Time of diagnosis						<0.01
Phase 1 (20 January to 19 November 2020)	25 (0.9)	6	24	19	76	
Phase 2 (20 November 2020 to 6 July 2021)	342 (12.7)	52	15.2	290	84.8	
Phase 3 (7 July 2021 to 29 January 2021)	2316 (86.3)	526	22.7	1790	77.3	
Symptom recognition on workdays vs. nonworkdays						
Weekday	1946 (72.5)	454	23.3	1492	76.7	
Public holiday, weekend	737 (27.5)	130	17.6	607	82.4	
Occupation group						<0.01
White collar	567 (21.1)	147	25.9	420	74.1	
Pink collar	434 (16.2)	69	15.9	365	84.1	
Blue collar	533 (19.9)	106	19.9	427	80.1	
Student	423(15.8)	123	29.1	300	70.9	
Economically inactive population	1149 (42.8)	262	22.8	887	77.2	
Vaccination						0.88
Unvaccinated	1228 (45.8)	263	21.4	965	78.6	
Partially vaccinated	258 (9.6)	55	21.3	203	78.7	
Fully vaccinated	1197 (44.6)	266	22.2	931	77.8	

^a^ Column percent; ^b^ row percent.

**Table 3 ijerph-21-00641-t003:** Delayed diagnosis (DD) by symptom.

Variable	Total N (%) ^a^	Tested within 24 h of Symptom Onset N% ^b^		DD (≥48 h after Symptom Onset) N% ^b^		*p*-Value
Total number of patients	2683 (100)	584	21.8	2099	78.2	
Number of symptoms						<0.01
1	631 (23.5)	158	25	473	75	
2	826 (30.8)	204	24.7	622	75.3	
3	582 (21.7)	106	18.2	476	81.8	
≥4	644 (24)	116	18	528	82	
Symptom ^c^						
Fever	928 (34.6)	271	29.2	657	70.8	<0.001
Chills	671 (25)	131	19.5	540	80.5	0.1
Cough	1273 (47.4)	227	17.8	1046	82.2	<0.001
Sputum	659 (24.6)	121	18.4	538	81.6	0.015
Difficulty breathing	72 (2.7)	10	13.9	62	86.1	0.1
Chest pain	25 (0.9)	1	4	24	96	0.031
Loss of consciousness	2 (0.1)	1	50	1	50	0.39 *
Cyanosis	1 (0)	0	0	1	100	1.00 *
Sore throat	1111 (41.4)	245	22.1	866	77.9	0.76
Headache	688 (25.6)	152	22.1	536	77.9	0.81
Myalgia	834 (31.1)	148	17.7	686	82.3	0.0007
Runny nose, nasal congestion	443 (16.5)	75	16.9	368	83.1	0.007
Fatigue	27 (1)	5	18.5	22	81.5	0.68
Diarrhea	40 (1.5)	7	17.5	33	82.5	0.51
Vomiting	20 (0.7)	3	15	17	85	0.59 *
Anosmia/ageusia	258 (9.6)	29	11.2	229	88.8	<0.001
Abdominal pain	4 (0.1)	2	50	2	50	0.21 *
Dizziness	30 (1.1)	5	16.7	25	83.3	0.5
Loss of appetite	8 (0.3)	1	12.5	7	87.5	1.00 *
Others	22 (0.8)	6	27.3	16	72.7	0.60 *

* The results of the Fisher’s exact test. ^a^ Column percent. ^b^ Row percent. ^c^ Adjusted for significance at a *p*-value ≤ 0.0025 according to the Bonferroni correction.

**Table 4 ijerph-21-00641-t004:** Delayed diagnosis (DD) by underlying disease.

Variable	Total N (%) ^a^	Tested within 24 h of Symptom Onset N% ^b^		DD (≥48 h after Symptom Onset) N% ^b^		*p*-Value
Total	2683 (100)	584	21.8	2099	78.2	
Underlying disease						
Present	794 (29.6)	136	17.1	658	82.9	<0.01
High risk	371 (13.8)	73	19.7	298	80.3	0.29
High-risk pre-existing conditions ^c^						
Diabetes	203 (7.6)	35	17.2	168	82.8	0.1
Cancer	34 (1.3)	6	17.6	28	82.4	0.56
Kidney dialysis	8 (0.3)	2	25	6	75	0.69 *
Heart disease	65 (2.4)	13	20	52	80	0.73
Cerebrovascular disease	31 (1.2)	10	32.3	21	67.7	0.15
Asthma	37 (1.4)	11	29.7	26	70.3	0.24
Pulmonary disease	25 (0.9)	4	16	21	84	0.48
Liver disease	6 (0.2)	0	0	6	100	0.35 *
Mental illness	5 (0.2)	2	40	3	60	1.00 *
Dementia	6 (0.2)	1	16.7	5	83.3	1.00 *
Others	9 (0.3)	2	22.2	7	77.8	1.00 *
Other pre-existing conditions ^c^						
Hypertension	445 (16.6)	66	14.8	379	85.2	<0.001
Dyslipidemia	184 (6.9)	19	10.3	165	89.7	<0.001
Thyroid dysfunction	27 (1)	9	33.3	18	66.7	0.14
Others	141 (5.3)	27	19.1	114	80.9	0.44

* The results of the Fisher’s exact test. ^a^ Column percent. ^b^ Row percent. ^c^ Adjusted for significance at a *p*-value ≤ 0.0033 according to the Bonferroni correction.

**Table 5 ijerph-21-00641-t005:** Results of the multivariate logistic regression analysis.

Variable	Total N	DD N (%)	Crude OR (95% CI)	aOR (95% CI) ^a^
Age				
0–19	348	229 (65.8)	Reference	Reference
20–39	931	701 (75.3)	1.58 (1.21–2.07)	2.03 (1.39–2.95)
40–64	966	797 (82.5)	2.45 (1.86–3.23)	3.12 (2.04–4.76)
≥65	438	372 (84.9)	2.93 (2.08–4.13)	3.6 (2.21–5.88)
Medical service area				
Chuncheon	904	683 (75.6)	Reference	Reference
Sokcho	360	293 (81.4)	1.42 (1.04–1.92)	1.3 (0.95–1.78)
Gangneung	359	265 (73.8)	0.91 (0.69–1.21)	0.83 (0.62–1.1)
Donghae	205	162 (79)	1.22 (0.84–1.76)	1.19 (0.82–1.74)
Yeongwol	111	92 (82.9)	1.57 (0.93–2.63)	1.41 (0.83–2.38)
Wonju	744	604 (81.2)	1.4 (1.1–1.77)	1.38 (1.08–1.76)
Time of diagnosis				
Phase 1 (20 January to 19 November 2020)	25	19 (76)	0.93 (0.37–2.34)	0.69 (0.27–1.79)
Phase 2 (20 November 2020 to 6 July 2021)	342	290 (84.8)	1.64 (1.2–2.24)	1.57 (1.14–2.15)
Phase 3 (7 July 2021 to 29 January 2021)	2316	1790 (77.3)	Reference	Reference
Symptom recognition on workdays vs. nonworkdays				
Weekday	1946	1492 (76.7)	Reference	Reference
Public holiday, weekend	737	607 (82.4)	1.42 (1.14–1.76)	1.41 (1.13–1.76)
Occupation group				
White collar	567	420 (74.1)	Reference	Reference
Pink collar	434	365 (84.1)	1.85 (1.35–2.55)	1.84 (1.33–2.54)
Blue collar	533	427 (80.1)	1.41 (1.06–1.87)	1.43 (1.07–1.91)
Student	423	300 (70.9)	0.85 (0.64–1.13)	1.92 (1.28–2.88)
Economically inactive population	726	587 (80.9)	1.48 (1.14–1.92)	1.44 (1.08–1.92)
Underlying disease				
Absent	1889	1441 (76.3)	Reference	Reference
Present	794	658 (82.9)	1.50 (1.22–1.86)	1.06 (0.82–1.36)

aOR: adjusted odds ratio; CI: confidence interval. ^a^ Adjusted variables: age, medical service area, time of diagnosis, symptom recognition on workdays vs. nonworkdays, and underlying disease (absent/present).

## Data Availability

The data that support the findings of the present study are available on request from the corresponding author. The data are not publicly available due to privacy and legal restrictions.

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
