# Peer review of "Influencing Factors of Delayed Diagnosis of COVID-19 in Gangwon, South Korea"

_ijerph, 2024, doi:10.3390/ijerph21050641_

Round 1
Reviewer 1 Report
Comments and Suggestions for Authors
Materials and Methods
Please provide a reference for the statement in the first sentence of the second paragraph.
You reported the finding for aOR which was supposed to be from multivariate analysis, however, the data analysis section did not state the methodology for achieving this analysis.
Author Response
Thank you for your review. As you suggested, we have added the following reference to the second paragraph of the Introduction, first sentence:
- Jeong, G.H.; Lee, H.J.; Lee, J.; Lee, J.Y.; Lee, K.H.; Han, Y.J.; Yoon, S.; Ryu, S.; Kim, D.K.; Park, M.B.; et al. Effective Control of COVID-19 in South Korea: Cross-Sectional Study of Epidemiological Data. J. Med. Internet Res. 2020, 22, e22103, doi:10.2196/22103.
Regarding the description of the multivariate analysis, it has been outlined in the “Data Analysis” section as follows. However, if any further clarification is needed, please let us know, and we will provide additional details. Thank you.
"With DD defined as a period exceeding 2 or more days from symptom recognition to COVID-19 diagnosis, multivariate logistic regression analysis was performed to examine the association of DD with the variables, including those that showed a p-value <0.2 in the univariate analysis, namely age, the medical service areas, time of diagnosis, symptom recognition on workdays vs. non-workdays, occupational groups, and underlying medical conditions."
Reviewer 2 Report
Comments and Suggestions for Authors
I thank the authors for giving me the opportunity to read this important and interesting work. The authors want to highlight the factors that were the causes of the diagnostic delay of COVID-19 in Gangwon, South Korea.
Although the article is clear, it requires a greater extension of the introduction and bibliography.
Author Response
Thank you for your review and comments. We have extended the Introduction section as follow and added references:
Early-stage COVID-19 patients are known for their ability to transmit the virus despite exhibiting mild symptoms [2]. It is estimated that, under normal circumstances, an infected individual may transmit the virus to an average of 2–4 others [3]. Therefore, early diagnosis of symptomatic COVID-19 cases has been recognized as imperative to mitigate its spread. Moreover, timely diagnosis of COVID-19 is essential for expedited patient management and improved health outcomes. Notably, previous cohort study findings have shown that even after adjusting for various confounding variables, a delay of more than five days from symptom onset to confirmed diagnosis was associated with a 69% increase in the likelihood of patients progressing to severe illness [4].
Reviewer 3 Report
Comments and Suggestions for Authors
Dear Editor,
The manuscript entitled "Influencing factors of delayed diagnosis of COVID-19 in Gangwon, South Korea" focuses on the epidemiological and social determinants that may increase the time between the onset of Covid-19 symptoms and testing by the public health service in Korea, based on the national case database between 2022 and 2022. The article is well-written and presents results consistent with the proposed methodology and the literature. Diagnosis time was a key factor during the pandemic. The laboratory surveillance of Covid-19 and its capacities that were established will certainly be a legacy for future health emergencies, justifying manuscripts of this nature.
Below are some major issues:
P2L58 – Authors should specify the difference between "public health center" and "public medical center".
P2L68 – Considering that PCR testing adopted during the Covid-19 epidemics was real-time with Ct values, authors should use the designation of RT-PCR or real-time RT-PCR.
P3L110 – By "imported case," do the authors mean from another country or another county (city)?
P3L122 - An analysis of the interval between the test date (the date of specimen collection) – Was testing always conducted on the same day as the collection? In other words, were swabs collected at residences or testing sites and sent to laboratories for Covid-19 detection on the same day? Or could the swabs be stored for processing on subsequent days while maintaining the original collection date? Perhaps explaining this process better to clarify that the entire process occurred on the same day.
- Where have the swabs been processed? RNA extraction and RT-PCR?
A major concern – Since the authors previously identified that 98% of cases were diagnosed within 24 hours of testing and, consequently, assumed that the outcome DD would be defined as a period exceeding 2 or more days from symptom recognition to COVID-19 diagnosis, could they be inducing information bias here? Instead, could they use the diagnostic time recommended by manuals and health authorities in Korea? For example, in several countries, a time of 48 or 72 hours has been determined.
P4L158 – Why was the time of testing categorized based on epidemic phases? Is there any indication or reference that there was an improvement in diagnosis time between periods?
P4L181 – Considering the high-risk conditions, are these conditions updated in today's protocols? Is reference 12 updated?
Tab 5. If DD was not significantly associated with underlying medical conditions, why has it been included in the final adjusted model?
P10L335 – Could the protective factor for DD in phase relative to phase 3 be due to the availability of commercial rapid tests available in pharmacies, leading to delayed testing?
Author Response
We sincerely appreciate your thoughtful review and valuable recommendations. In response to your insightful feedback, we have meticulously revised the content of the article. Below, we have provided detailed responses (black color) to each of the points you raised (red color).
P2L58 – Authors should specify the difference between "public health center" and "public medical center".
→ Authors’ response:
As you commented, we have added the following content to specify the difference between a "public health center" and a "public medical center".
Gangwon is composed of 18 Si (cities) and Gun (counties), each hosting a public health center. Public health centers, established with the purpose of optimizing health admin-istration and effectively promoting health policies, serve as pivotal institutions within the cities and counties of Korea [11]. They also play a crucial role in local health initiatives, encompassing functions of both public health and primary care, with a notable emphasis on the implementation of public health management strategies [12,13]. In specific regions characterized by limited medical resources, public health centers function as comprehen-sive healthcare facilities, equipped with both outpatient and inpatient treatment capabili-ties, alongside their local public health management functions [14]. Among these 18 pub-lic health centers in Gangwon, two are designated as public medical centers, providing essential emergency care and medical services.
P2L68 – Considering that PCR testing adopted during the Covid-19 epidemics was real-time with Ct values, authors should use the designation of RT-PCR or real-time RT-PCR.
→ Authors’ response:
As you recommended, we have revised the term "PCR" in the text to "real-time RT-PCR" or "RT-PCR" as per your instructions.
P3L110 – By "imported case," do the authors mean from another country or another county (city)?
→ Authors’ response:
The term "imported case" referred to cases imported from another country. Therefore, we have modified it to "imported case from abroad."
P3L122 - An analysis of the interval between the test date (the date of specimen collection) – Was testing always conducted on the same day as the collection? In other words, were swabs collected at residences or testing sites and sent to laboratories for Covid-19 detection on the same day? Or could the swabs be stored for processing on subsequent days while maintaining the original collection date? Perhaps explaining this process better to clarify that the entire process occurred on the same day.
- Where have the swabs been processed? RNA extraction and RT-PCR?
→ Authors’ response:
To clarify that the entire process occurred on the same day, the following sentence was added to "2. Materials and Methods - Setting" section with a reference.
Swabs obtained were conveyed to the Gangwon State Institute of Health and Environment or private research institutes within Gangwon, for RNA extraction and subsequent RT-PCR analysis. Commencing from January 24, 2020 and continuing throughout the study period, the Gangwon State Institute of Health and Environment maintained a 24-hour emergency response system, enabling uninterrupted COVID-19 diagnostic testing. RT-PCR-based COVID-19 testing operated under the guiding principle of analyzing specimens on the day of collection to ensure expedited results. Consequently, most specimens underwent immediate transportation to the laboratory upon collection [16].
A major concern – Since the authors previously identified that 98% of cases were diagnosed within 24 hours of testing and, consequently, assumed that the outcome DD would be defined as a period exceeding 2 or more days from symptom recognition to COVID-19 diagnosis, could they be inducing information bias here? Instead, could they use the diagnostic time recommended by manuals and health authorities in Korea? For example, in several countries, a time of 48 or 72 hours has been determined.
→ Authors’ response:
The South Korean government consistently promoted and encouraged individuals to undergo COVID-19 testing "immediately" upon the manifestation of symptoms. Considering the time required for the transportation of COVID-19 test samples and the typical duration of approximately six hours for COVID-19 RT-PCR testing, samples collected during late evenings or at night are more likely to yield results the following day. Therefore, we determined that defining a delay in diagnosis as a difference of more than two days from symptom onset was most appropriate, believing that this threshold would minimize the potential for information bias. With this consideration in mind, we have added the following content to the "2. Materials and Methods - Variables" section.
Considering the WHO and governmental recommendations for immediate testing upon symptom onset, testing on the day symptoms appear would be ideal. However, when considering the time required for the transportation of COVID-19 test samples and the typical duration of approximately six hours for COVID-19 RT-PCR testing, samples collected during late evenings or at night are more likely to yield results the following day [17]. Additionally, since the majority of cases had a TTD within 1 day, classifying timely testing as a TTD of 0 or 1 day, and a TTD of 2 or more days as DD, was deemed appropriate to minimize the potential for information bias.
P4L158 – Why was the time of testing categorized based on epidemic phases? Is there any indication or reference that there was an improvement in diagnosis time between periods?
→ Authors’ response:
Since February 7, 2020, following the introduction of real-time RT-PCR diagnostic kits in Korea, the testing time has been significantly reduced from 24 hours to just 6 hours. Throughout the research period, Korea exclusively relied on the RT-PCR method for diagnosing COVID-19. Rapid antigen tests were not employed for confirming the diagnosis of COVID-19 patients during our study period. Therefore, we are confident in our ability to classify the testing time according to the epidemic phase in this study.
P4L181 – Considering the high-risk conditions, are these conditions updated in today's protocols? Is reference 12 updated?
→ Authors’ response:
Thank you for informing us about the updated protocols. We have confirmed that this reference was updated on April 12, 2024. We compared the Higher Risk condition section of the document we referenced in our paper, which was the February 9, 2023 version, with the newly updated document. We found no changes in the Higher Risk condition section. However, we still updated our reference using the most up-to-date protocol.
Tab 5. If DD was not significantly associated with underlying medical conditions, why has it been included in the final adjusted model?
→ Authors’ response:
As can be seen in Table 4, the presence of underlying disease was significantly associated with an increase in DD. Therefore, we included the presence of underlying disease in the final adjusted model. However, high-risk underlying medical conditions did not show statistically significant association with DD, so they were not included in the final adjusted model.
P10L335 – Could the protective factor for DD in phase relative to phase 3 be due to the availability of commercial rapid tests available in pharmacies, leading to delayed testing?
→ Authors’ response:
During the study period, only cases of COVID-19 confirmed by RT-PCR were recognized in Korea. ,Therefore, anyone desiring a COVID-19 test could receive the RT-PCR test for free of charge. However, with the surge in cases due to the Omicron variant, guidelines were revised on February 10, 2022. From then on, only individuals aged 60 and above, those with epidemiological associations, and those deemed necessary for COVID-19 testing based on medical advice were eligible for free COVID-19 PCR tests, excluding those who tested positive in rapid antigen tests or emergency screening tests. Since this study only includes cases confirmed before February 10, 2022, we believe that the concern raised by the reviewer regarding the availability of commercial rapid tests leading to delayed testing would not be relevant in our study.